# Lovastatin, an Up-Regulator of Low-Density Lipoprotein Receptor, Enhances Follicular Development in Mouse Ovaries

**DOI:** 10.3390/ijms24087263

**Published:** 2023-04-14

**Authors:** Yu Jin Kim, Yong Il Cho, JuYi Jang, Yun Dong Koo, Sung Woon Park, Jae Ho Lee

**Affiliations:** 1CHA Fertility Center Seoul Station, Seoul 04637, Republic of Korea; 2Wonju Severance Christian Hospital, Wonju 22070, Republic of Korea; 3Department of Biomedical Sciences, CHA University, Pocheon 11160, Republic of Korea

**Keywords:** LDLR, lovastatin, follicular development, oocytes, in vitro maturation

## Abstract

Ovarian aging hampers in vitro fertilization in assisted reproductive medicine and has no cure. Lipoprotein metabolism is associated with ovarian aging. It remains unclear how to overcome poor follicular development with aging. Upregulation of the low-density lipoprotein receptor (LDLR) enhances oogenesis and follicular development in mouse ovaries. This study investigated whether upregulation of LDLR expression using lovastatin enhances ovarian activity in mice. We performed superovulation using a hormone and used lovastatin to upregulate LDLR. We histologically analyzed the functional activity of lovastatin-treated ovaries and investigated gene and protein expression of follicular development markers, using RT-qPCR and Western blotting. Histological analysis showed that lovastatin significantly increased the numbers of antral follicles and ovulated oocytes per ovary. The in vitro maturation rate was 10% higher for lovastatin-treated ovaries than for control ovaries. Relative LDLR expression was 40% higher in lovastatin-treated ovaries than in control ovaries. Lovastatin significantly increased steroidogenesis in ovaries and promoted the expression of follicular development marker genes such as anti-Mullerian hormone, Oct3/4, Nanog, and Sox2. In conclusion, lovastatin enhanced ovarian activity throughout follicular development. Therefore, we suggest that upregulation of LDLR may help to improve follicular development in clinical settings. Modulation of lipoprotein metabolism can be used with assisted reproductive technologies to overcome ovarian aging.

## 1. Introduction

In assisted reproductive medicine, age is a major factor determining whether a successful pregnancy is achieved following in vitro fertilization (IVF) [1,2]. The average age of people getting married has increased, and the mean age of infertile women has dramatically increased compared with the past decade [3]. Therefore, studies have investigated how to overcome female infertility related to advanced age [4,5,6]. However, no curative treatment is available for infertility related to advanced age, which manifests as a reduced competence of oocytes and embryos in assisted reproductive technologies [7].

Oogenesis and follicular development are highly complex processes that are regulated by multiple factors and involve the gonadotropin-independent phase for development of primary oocytes, and the gonadotrophin-dependent phase for development and maturation of antral follicles [8,9]. 

The quality and quantity of oocytes are associated with pregnancy ratios following IVF in assisted reproductive medicine. Among women of advanced age undergoing IVF, those with a history of successful live births may have an adequate quality and quantity of oocytes in contrast with those who have experienced failed pregnancies [10]. Unfortunately, no agents have been identified that can restore the quality and quantity of oocytes in females of advanced age. 

Metabolism of lipoproteins during steroidogenesis in oogenesis and follicular development for ovulation is highly complex and involves several factors depending on the follicular stage [11,12,13,14]. The starting point of steroidogenesis is cholesterol influx from extracellular fluid dependent on low-density lipoprotein receptor (LDLR) signaling [13]. Recently, our group reported that LDLR is associated with female infertility [13]. The LDLR gene family comprises cell surface proteins that are involved in receptor-mediated endocytosis of specific ligands. These proteins are normally bound to low-density lipoprotein/cholesterol at the cell membrane and are taken up by the cell. Next-generation sequencing revealed that factors such as LDLR are upregulated in cumulus cells of pregnant women [13]. LDLR is the key upregulated factor responsible for successful pregnancies in both old and young women. The major role of LDLR is to import precursors for steroids such as estradiol (E2) and progesterone, which are involved in gonadotropin-dependent oogenesis, ovulation, and implantation during the luteal phase. StAR is involved in cholesterol transfer from the cytoplasm to mitochondria for steroidogenesis [15]. Therefore, LDLR and StAR are key factors for successful pregnancies in the context of steroid metabolism. Even older patients exhibit highly activated steroidogenesis, reflected by upregulation of LDLR and StAR [13]. These factors improve embryo quality and blastocyst ratios and help to achieve a successful clinical outcome. During steroidogenesis in ovaries, LDLR is the starting point of E2 synthesis in human cumulus cells. We previously found that LDLR is important for the activation of aged ovaries to produce good-quality oocytes and embryos. Upregulation of LDLR is associated with clinical pregnancy [13]. Specifically, the pregnancy group shows significantly higher LDLR expression than the nonpregnancy group in cumulus cells. 

Based on our previous study, we tried to upregulate LDLR with drugs such as statins. Statins have been used to clinically treat lipoprotein disorders. Several types of statins are used to enhance LDLR expression in people with such disorders. Some studies have shown that statins have the potential to recover steroidogenesis in ovaries. However, until now statins did not apply to the enhancement of oogenesis and follicular development in the reproductive sciences. Moreover, the effect of upregulated LDLR on female infertility is unclear. Additionally, agents that upregulate LDLR have not been applied to enhance ovarian function in infertile women of advanced age. 

Therefore, we investigated whether lovastatin enhances LDLR expression in mouse ovaries and analyzed oogenesis and follicular development in lovastatin-treated mouse ovaries.

## 2. Results

### 2.1. Viability of Lovastatin-Treated Cells

HEK293 cells were incubated for 24 h in Dulbecco’s Modified Eagle Medium (DMEM) containing 10% fetal bovine serum (FBS) supplemented with 0.01, 0.1, 1, 10, or 100 µM lovastatin. Treatment with lovastatin at a concentration of 1 µM or lower did not significantly affect cell viability compared with the control group. However, treatment with lovastatin at a concentration of 10 µM or higher significantly decreased cell viability (Appendix A).

### 2.2. Body Weight Changes of Lovastatin-Treated Mice

After intraperitoneal injection of mice with lovastatin for 10 days, we observed changes in the body weights of the mice. We set the day when the injection started as day 1 (postnatal 28 days). Lovastatin-treated mice gained less body weight than control mice from day 3 to 10 (Figure 1).

### 2.3. Lovastatin Treatment Enhances Follicular Development in Hormone-Stimulated Mouse Ovaries

We evaluated the effect of lovastatin treatment on mouse ovaries stimulated with pregnant mare serum gonadotropin in vivo. Figure 2A shows histological images of mouse ovarian tissues stained with hematoxylin and eosin following lovastatin treatment and hormone stimulation. Histological data reveal significantly increased antral follicles and ovulated follicles in the Lovastatin-treated ovary. Additionally, we counted the number of oocytes per ovary. More germinal vesicle (GV) oocytes were collected from lovastatin-treated ovaries than from control ovaries (Figure 2B). We analyzed in vitro maturation ratios after the culture of immature oocytes of each group. We determined the maturation percentage of oocytes with germinal vesicle breakdown (GVBD) and metaphase II (MII) formation. Lovastatin-treated oocytes revealed that the GVBD phase was lower than in the control group, but the high percentage of oocytes in the MII phase was higher (Table 1). In summary, lovastatin enhanced follicular development in mouse ovaries. 

### 2.4. Expression of Primordial Oogenesis Factors and Ovarian Function Markers Is Increased in Lovastatin-Treated Mouse Ovaries

We compared the expression of steroidogenesis and ovarian function markers between control and lovastatin-treated ovaries. RT-PCR and real-time qPCR revealed that the mRNA expression of ovarian function markers such as StAR, anti-Mullerian hormone (AMH), growth differentiation factor 9 (GDF9), and bone morphogenetic protein 15 (BMP15) was significantly higher in lovastatin-treated ovaries than in control ovaries. AMH had no significant difference between the control and experiment groups. Lovastatin treatment also increased LDLR expression (Figure 3A,B). Protein expression of LDLR, StAR, Oct3/4, and Nanog was markedly higher in lovastatin-treated ovaries than in control ovaries (Figure 3C). 

### 2.5. Immunohistological Analysis Using an Anti-LDLR Antibody

Immunohistological analysis revealed that the staining of LDLR was markedly higher in lovastatin-treated ovaries than in control ovaries. In particular, cumulus cells were immunostained more with an anti-LDLR antibody in lovastatin-treated ovaries than in control ovaries (Figure 4). 

## 3. Discussion

In this study, we investigated the effect of lovastatin, an up-regulator of LDLR, on follicular development of oocytes in mouse ovaries. Therefore, lovastatin treatment enhanced both the gonadotrophin-independent and -dependent phases on hormone hyperstimulation as well as the gene and protein expression of follicular development and ovarian function markers.

Lovastatin enhanced the gonadotropin-independent phase via the LDLR signaling pathway. It upregulated Oct3/4, Nanog, and Sox2, which are expressed in primary oocytes, via LDLR signaling. Moreover, lovastatin treatment enhanced the activity of primordial germ cells. Oct3/4, Nanog, and Sox2 are markers of primordial germ cells and regulate the proliferation of these cells and their differentiation into preantral follicles [8]. Oct3/4 and Nanog are involved in the proliferation and differentiation of human primary oocytes in the ovary [9]. The pluripotency gene Oct3/4 is important for the survival of primordial germ cells in females of advanced age [16]. Lovastatin may facilitate controlled ovarian stimulation by upregulating Oct3/4 and Nanog and thereby enhancing the survival of primordial germ cells in ovaries during IVF in assisted reproductive medicine.

We previously reported the mechanism by which LDLR signaling enhances follicular development in ovaries by performing whole-transcriptome next-generation sequencing [13]. This study revealed that the LDLR gene and genes involved in steroidogenesis and follicular development were upregulated in lovastatin-treated ovaries. Lovastatin upregulated several oogenesis and follicular development markers, such as AMH, GDF9, and BMP15.

AMH is a critical marker of ovarian activity and is utilized as an indicator of human ovarian function in clinical settings [17]. Lovastatin upregulated AMH through the LDLR signaling pathway. GDF9 and BMP15 are expressed in cumulus cells. They are major regulators of the functional activity of cumulus cells and are involved in oogenesis and follicular development. GDF9 and BMP15 are associated with premature ovarian failure [18]. StAR is important for the initial phase of steroidogenesis in mitochondria. Cholesterol is transported into cumulus cells and used to synthesize sex hormones at the mitochondrial membrane [19].

Steroidogenesis in ovaries is important for cumulus–oocyte complexes and follicular granulosa cells (Figure 5). Therefore, steroidogenesis occurs in follicles depending on the developmental stage and gonadotropin status. In preantral follicles, E2 and progesterone are produced by cumulus and granulosa cells in the ovary, as indicated by the high expression of 3-β-hydroxy-steroid-dehydrogenase and cholesterol side-chain cleavage enzyme, and androgens are synthesized and converted into 17β-estradiol by granulosa aromatase P450 [20]. The biosynthesis of steroids in granulosa and cumulus cells in the ovary is associated with ovarian function such as follicular development, oocyte maturation, and oocyte quality [6,21,22]. Lovastatin affects the gonadotropin-independent and -dependent phases of follicular development via LDLR signaling.

Despite many studies, an efficient stimulation protocol for patients of advanced age is lacking. Most treatments that improve pregnancy rates are not recommended for females of advanced age, and overcoming female aging remains one of the greatest challenges in reproductive medicine. We suggest that an LDLR up-regulator, such as lovastatin, could be further investigated to enhance follicular development and, thus, facilitate IVF in infertile female patients of advanced age in clinical trials.

## 4. Materials and Methods

### 4.1. Materials

Lovastatin was purchased from Sigma (PHR1285-1G; St. Louis, MO, USA).

### 4.2. Animals

Outbred white ICR mice (6–8 weeks old, 20–25 g in weight) were purchased from Oriental Bio (Seongnam, Republic of Korea). All procedures for animal care complied with the regulations of the Institutional Animal Care and Use Committee of Sahmyook University (Seoul, Republic of Korea). Mice were maintained under standard conditions at 22 ± 3 °C, 60% humidity and 12 h light/dark cycle with regular feeding.

### 4.3. Determination of the Effect of Lovastatin on Ovaries In Vivo

Lovastatin-treated mice (*n* = 30) and control mice (*n* = 30) were analyzed. Lovastatin-treated mice were injected every day with 0.5 mg/kg lovastatin for 10 days. Then we performed superovulation. To induce superovulation, 75 IU of pregnant mare serum gonadotropin was injected 10 days later. Mice were euthanized by the inhalation of CO_2_ gas in a chamber. Both ovaries were collected from the mouse and separated by halving into groups. Using half of an ovary, Oocytes were harvested from the intact ovary tissue by puncturing the antral follicle (500 μm) with a 28 G microneedle and by observing under a stereomicroscope (N88; Nikon, Tokyo, Japan). The other half of the ovary was separated into three different groups. Another ovary half was fixed with 4% paraformaldehyde for histological analysis. The other ovary half was stored at −80 °C for analysis of oogenesis by RT-PCR and Western blotting.

Therefore, we collected immature oocytes from each of the 15 control and experimental ovaries. Immature oocytes were cultured with 5 μM milrinone (78415-72-2; Sigma-Aldrich, St. Louis, MO, USA) to ensure germinal vesicle (GV) stage synchronization. The milrinone reagent was removed by washing, and oocyte maturation from the GVBD to the MII stage was induced using SAGE in vitro maturation media (IVM) (CooperSurgical^®^; ART-1600-8; Ballerup, Denmark). The SAGE™ IVM medium was supplemented with a serum protein substitute (SPS) and 75 IU of follicle-stimulating hormone (FSH) and luteinizing hormone (LH). The oocytes were incubated for 15 h in a humidified incubator (Heracell 240; Heraeus, Hanau, Germany) with 5% CO_2_ at 37 °C, for oocyte maturation to the MII stage, and images were captured after 4 and 15 h. Then, we analyzed the immature ratios of GV oocytes on each sample. We repeated this set of experiments thrice for statistical analysis.

### 4.4. Histological Analysis and Immunohistochemistry

Mouse ovaries were imaged using light and confocal microscopy. Ovaries were harvested and fixed with 4% paraformaldehyde for 10 min at room temperature. Hematoxylin and eosin staining was performed using routine protocols for ovarian histological analysis.

For immunohistochemical staining, ovarian tissue-containing paraffin sections on glass slides were deparaffinized by incubation in a xylene series ranging from 100% to 75% for 10 min each, and then rehydrated by incubation in an ethanol series ranging from 100% to 75% for 10 min each at room temperature. For antigen retrieval, the tissue sections were boiled in 0.01 M citrate buffer (pH 6.0) for 20 min in an autoclavable jar. After permeabilization with PBS containing 0.1% Triton X-100 for 20 min at room temperature, the sections were washed three times with fresh PBS. Subsequently, the sections were incubated overnight at 4 °C with a rabbit anti-LDLR primary antibody (MA5-32075; Invitrogen, Waltham, MA, USA) diluted 1:100 in PBS containing 0.03% bovine serum albumin. For DAB staining, a horseradish peroxidase-conjugated secondary antibody was applied to tissue sections, and counterstaining with hematoxylin was performed for 5 min at room temperature. Then, slides were mounted using mounting media and covered with coverslips. Images of DAB staining were acquired using an inverted light microscope (Eclipse Ti2; Nikon, Tokyo, Japan) equipped with a camera (DS-Ri2; Nikon, Tokyo, Japan) and imaging software (NIS-Elements ver. 4.4.; Nikon, Tokyo, Japan).

### 4.5. Gene Expression Analysis

Total RNA was purified from ovaries using TRIzol solution (15596026; Invitrogen, Carlsbad, CA, USA) according to the manufacturer’s procedures. Isolated total RNA was transcribed into cDNA using AccuPower^®^ CycleScript RT PreMix (K-2050; Bioneer, Daejeon, Republic of Korea) and amplified with AccuPower^®^ Taq PCR Premix (K-2602; Bioneer, Daejeon, Republic of Korea) using primer sets specific for mouse LDLR, StAR, AMH, GDF9, BMP15, Oct3/4, Nanog, Sox2, and β-actin (Appendix A). In total, 10 pmol/µL forward and reverse primers and 200 ng of template cDNA were added to AccuPower^®^ Taq PCR PreMix tubes, and then distilled water was added up to a total volume of 20 µL. The PCR cycling conditions were as follows: 1 min at 95 °C, followed by 30 cycles of denaturation for 30 s at 95 °C, annealing for 30 s at 60 °C, and extension for 50 s at 72 °C. Amplified PCR products were resolved on 1.5% agarose gels with Safeview™ FireRed (G926; Applied Biological Materials, Richmond, BC, Canada). Agarose gels were visualized under ultraviolet illumination using a gel documentation system (WSE-6100 LuminoGraph; ATTO, Tokyo, Japan).

### 4.6. Quantitative Real-Time PCR

Quantitative real-time PCR was performed with total RNA isolated from ovaries using TRIzol. The concentration of total RNA was measured using a spectrophotometer (Epoch Microplate Spectrophotometer; BioTek Instruments Inc., Winooski, VT, USA). Total RNA was reverse-transcribed into cDNA using AccuPower^®^ CycleScript RT PreMix (K-2050; Bioneer, Daejeon, Republic of Korea). The qPCR was performed using AccuPower GreenStar qPCR PreMix (K-6212; Bioneer, Daejeon, Republic of Korea) on a spectrofluorometric thermal cycler (CFX96 Touch Real-Time PCR Detection System; Bio-Rad, Hercules, CA, USA). The reaction contained 10 pmol/µL forward and reverse primers, 200 ng of template cDNA, and AccuPower GreenStar qPCR PreMix. Distilled water was added up to a final volume of 20 µL. The PCR cycling conditions were as follows: 3 min at 95 °C, followed by 40 cycles of denaturation for 10 s at 95 °C, and annealing and extension for 20 s at 60 °C. Expression of each gene was normalized to that of β-actin. Triplicate samples were tested. All experiments were repeated three times for statistical analysis with CFX Maestro ver. 2.5 (Bio-Rad, Hercules, CA, USA).

### 4.7. Protein Expression Analysis

Lovastatin-treated and control ovaries were homogenized with 1.0 mL of protein extraction buffer (17081; iNtRON Biotechnology, Seongnam, Republic of Korea). The concentration of protein extracted was measured through a BCA assay (BCA0500; Biomax, Republic of Korea). The extracted protein samples were boiled with 4× Laemmli buffer (1610747; Bio-Rad), resolved by 8% SDS-PAGE, and transferred to nitrocellulose membranes (BR162-0112; Bio-Rad). The membranes were incubated for 1 h at room temperature in blocking buffer, which consisted of Tris-buffered saline supplemented with 0.5% Tween 20 and 5% bovine serum albumin, and then incubated with rabbit anti-LDLR (MA5-32075; Invitrogen, Waltham, MA, USA), rabbit anti-StAR (PA5-21687; Invitrogen, Waltham, MA, USA), rabbit anti-Oct3/4 (sc-9081; Santa Cruz Biotechnology, Dallas, TX, USA), mouse anti-Nanog (sc-134218; Santa Cruz Biotechnology), and mouse anti-β-actin (MA5-15739; Invitrogen, Waltham, MA, USA) primary antibodies overnight at 4 °C. After four washes for 5 min each with Tris-buffered saline containing 0.5% Tween 20, the membranes were incubated with horseradish peroxidase-conjugated goat anti-mouse (62-6520; Invitrogen, Waltham, MA, USA) and goat anti-rabbit (31460; Thermo Fisher, Waltham, MA, USA) secondary antibodies. Immunoreactive bands were visualized using enhanced chemiluminescence (Miracle-Star™, 16028, iNtRON Biotechnology, Seongnam, Republic of Korea). Images of the bands were acquired using a gel documentation system (LuminoGraph I, WSE-6100; ATTO, Tokyo, Japan).

### 4.8. Statistical Analysis

All experiments were repeated three times. Statistical analysis was performed using a *t*-test analysis of variance with SigmaPlot 12.5 software. The significance level was set at ** p* < 0.05 and *** *p* < 0.001.

## Figures and Tables

**Figure 1 ijms-24-07263-f001:**
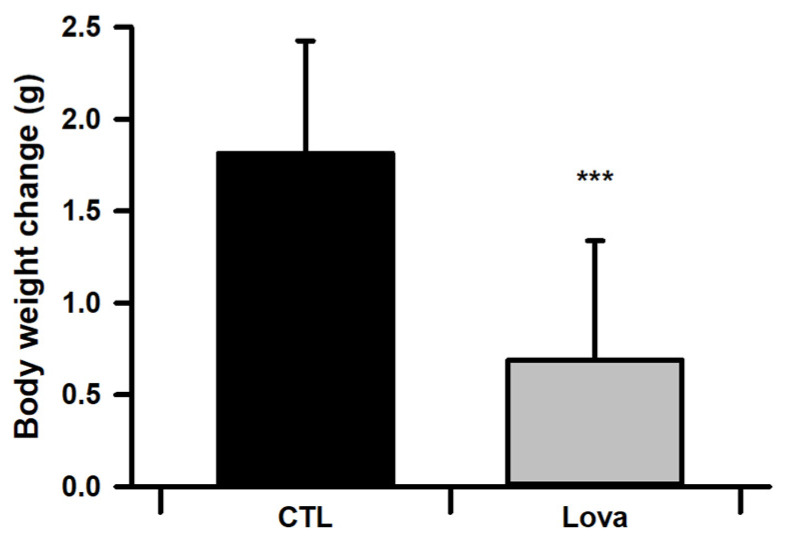
Average of body weight for control and lovastatin-treated mice from day 1 to 10 after Lovastatin injection (*n* = 30, *** *p* < 0.001).

**Figure 2 ijms-24-07263-f002:**
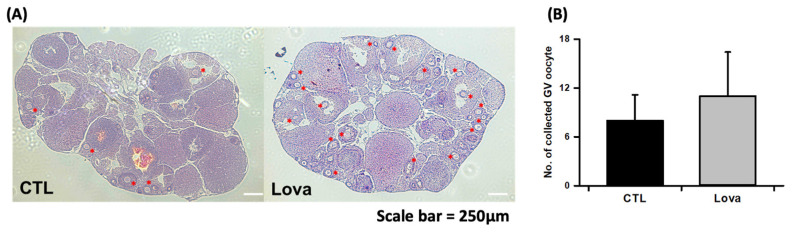
Characterization of lovastatin-treated ovaries. (**A**) Light microscopic images of hematoxylin and eosin-stained mouse ovarian tissue treated without (CTL) and with lovastatin (Lova). Asterisk is primary oocyte and follicles. Scale bar = 250 µm. (**B**) Number of GV oocytes collected from control and lovastatin-treated ovaries. Data are the mean ± standard error of the mean of four replicates. Control mice ovaries (*n* = 30) and Lovastatin-treated mice ovaries (*n* = 30).

**Figure 3 ijms-24-07263-f003:**
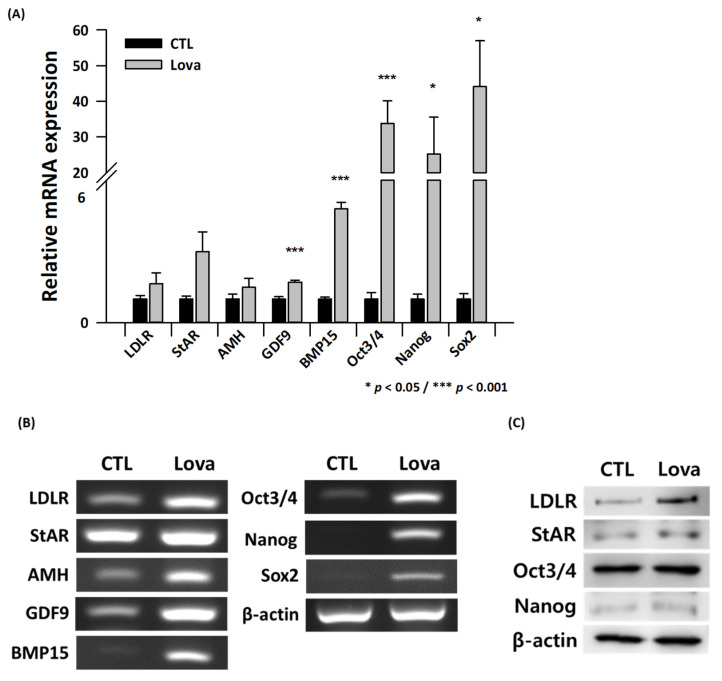
Expression of steroidogenesis and ovarian function markers in lovastatin-treated ovaries. (**A**) Quantitative real-time PCR analysis of LDLR, StAR, AMH, GDF9, BMP15, Oct3/4, Nanog, and Sox2 expression normalized against β-actin expression in control (CTL) and lovastatin-treated (Lova) ovaries. Data are the mean ± standard error of the mean of three replicates (* *p* < 0.05 and **** p* < 0.001 versus the control group) (*n* = 3). (**B**) RT-PCR analysis of control (CTL) and lovastatin-treated (Lova) ovaries. (**C**) Western blot analysis of control (CTL) and lovastatin-treated (Lova) ovaries.

**Figure 4 ijms-24-07263-f004:**
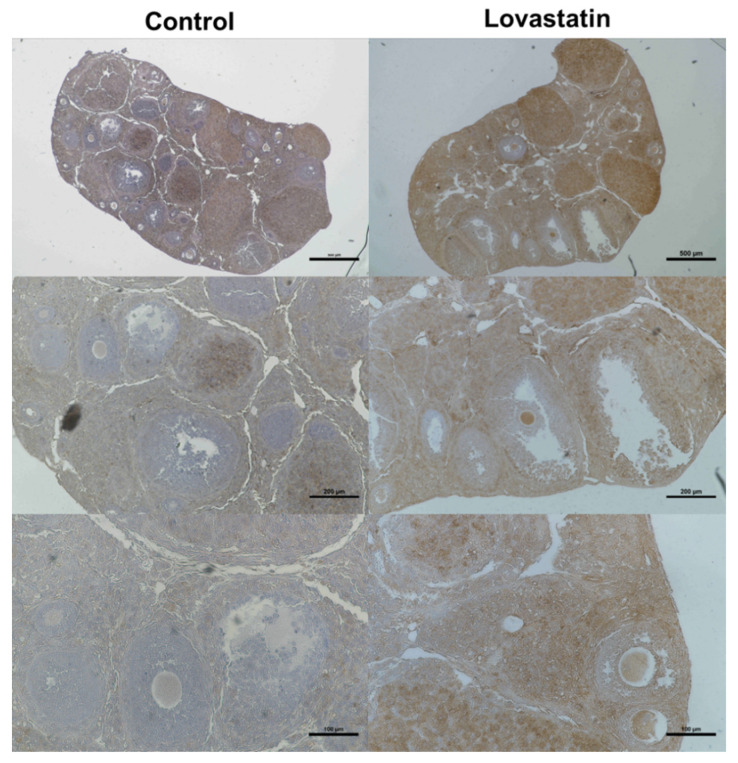
Immunohistochemical analysis of LDLR expression in control and lovastatin-treated ovaries. Right and left panels show lovastatin-treated and control ovaries, respectively, at different magnifications (top: 40×, middle: 100×, and bottom: 200×). Scale bars are 500, 200, and 100 µm in the top, middle, and bottom panels, respectively (*n* = 3).

**Figure 5 ijms-24-07263-f005:**
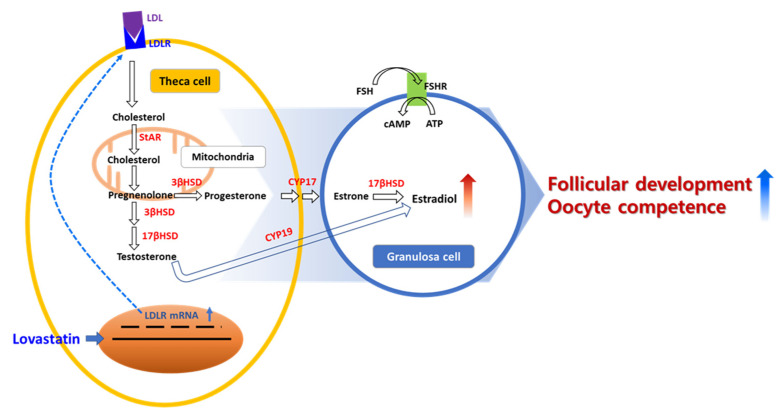
Scheme of LDLR activity signaling pathway for follicular development with lovastatin stimulates steroidogenesis in ovaries.

**Table 1 ijms-24-07263-t001:** Number of oocytes harvested from control and lovastatin-treated oocytes, and in vitro maturation percentages of GV oocytes into GVBD (germinal vesicle breakdown) and MII (metaphase II) phases.

	Total Collected GV Oocytes	Total Collected Degradation GV	GVBD (%)	MII (%)	Not Mature (%)
CTL	32	27	14 (44%)	13 (41%)	5 (16%)
Lova	44	21	12 (27%)	24 (55%)	7 (16%)

CTL: control, Lova: Lovastatin

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
