# Peer review of "Lovastatin, an Up-Regulator of Low-Density Lipoprotein Receptor, Enhances Follicular Development in Mouse Ovaries"

_ijms, 2023, doi:10.3390/ijms24087263_

Round 1

Reviewer 1 Report

The manuscript entitled Lovastatin, an up-regulator of low-density lipoprotein receptor, 2 enhances follicular development in mouse ovaries brings out very interesting points on fertility and aging and illuminates this increasing problem from applicable perspective.

The manuscript is delivered in a clear and precise manner. The methodology is sufficiently detailed, the results are presented in illustrative tables and figures, and conclusions are adequate and well understandable. Analyses and conclusions are of high interest for contemporary fertility industry.

I am delighted to suggest the publication of this article in present form.

Author Response

Reviewer 1#

Comments and Suggestions for Authors

The manuscript entitled Lovastatin, an up-regulator of low-density lipoprotein receptor, 2 enhances follicular development in mouse ovaries brings out very interesting points on fertility and aging and illuminates this increasing problem from applicable perspective.

The manuscript is delivered in a clear and precise manner. The methodology is sufficiently detailed, the results are presented in illustrative tables and figures, and conclusions are adequate and well understandable. Analyses and conclusions are of high interest for contemporary fertility industry.

I am delighted to suggest the publication of this article in present form.

  1. R) thank you very much, good consideration to my manuscript.

Reviewer 2 Report

The manuscript readably focuses on Lovastatin enhancing follicular development in mouse ovaries and its possible mechanism. The experimental design and scientific integrity of this research are quite appreciable. However, the major drawback of the manuscript is poor drafting and several results do not explain properly, which I believe can be improved. The following are more detailed comments.

The background description is confusing and not cohesive enough, please modify it!

English description should be further strengthened. Native English-speaking professionals should be asked to make modifications!

The discussion section should be further modified!

Lines 14-17 do not connect

Lines 28-30, this paper does not do any trial on assisted reproductive technology, so it cannot be described in this way

Lines 39-41 should be attributed to the end of the first paragraph

What is the correlation between lines 41-44 and 44-45? Disconnection of context

Please add references for lines 57 to 60

What's the difference between lines 59-60 and 64-65? Please delete or modify lines 59 to 60 LDLR and StAR are the key…

In lines 73 to 79, please add more background detailing the effects of statins on ovarian steroids

Lines 92 to 95, does the author notice the body weight ratio to ovary weight, or to reproductive fat? Please add

In lines 97-98, please add n=x and ***p < 0.001 in Figure 1

Figure 2A should indicate what are antral follicles and what are ovulated follicles

Why is the mean standard error not shown in Figure 2C? Is only one mouse counted?

FIG. 3B, FIG. 3C and FIG. 4 should be quantified so that the differences can be seen more directly

Please add n=X information to the notes for Figures 2, 3 and 4

In lines 172-177, whether Figure 5 is verified or conjectured, please add background information for further details

For lines 196-199, please add feeding conditions for mice

Line 227, 4.5. Gene expression analysis should include PCR and qPCR. I suggest combining them together or modifying the title 4.5

Lines 240-270, Please add quantitative analysis

Please add full blotting in supplementary material

The P value of the full text should be italicized

More attention should be paid to the superscript and subscript of the manuscript and annex text, such as lines 5 and 9 of the annex

Author Response

Reviewer 2#

Comments and Suggestions for Authors

The manuscript readably focuses on Lovastatin enhancing follicular development in mouse ovaries and its possible mechanism. The experimental design and scientific integrity of this research are quite appreciable. However, the major drawback of the manuscript is poor drafting and several results do not explain properly, which I believe can be improved. The following are more detailed comments.

The background description is confusing and not cohesive enough, please modify it!

English description should be further strengthened. Native English-speaking professionals should be asked to make modifications!

The discussion section should be further modified!

Lines 14-17 do not connect

  1. R) we have seperated the sentence

Lines 28-30, this paper does not do any trial on assisted reproductive technology, so it cannot be described in this way

  1. R) Thank you for your good comment. I’m re-writing the sentence and delete.

Lines 39-41 should be attributed to the end of the first paragraph

  1. R) Revise it.

What is the correlation between lines 41-44 and 44-45? Disconnection of context

  1. R) Revise it.

Please add references for lines 57 to 60

  1. R) Re-writing the part and updated on the sentence.

What's the difference between lines 59-60 and 64-65? Please delete or modify lines 59 to 60 LDLR and StAR are the key…

  1. R) Thank you for the good comment, we modified each sentence and re-writing following your comment

In lines 73 to 79, please add more background detailing the effects of statins on ovarian steroids

  1. R) we have been searching for statins in the ovarian steroids study. There is no reported effect of statins on the ovary like oogenesis and follicle development.

Lines 92 to 95, does the author notice the body weight ratio to ovary weight, or to reproductive fat? Please add

  1. R) Body weight is just body weight. We didn’t analyze ovary weight.

In lines 97-98, please add n=x and ***p < 0.001 in Figure 1

  1. R) Added it

Figure 2A should indicate what are antral follicles and what are ovulated follicles

  1. R) we indicated in the image by asterisk markers at primary oocytes and follicles.

Why is the mean standard error not shown in Figure 2C? Is only one mouse counted?

  1. R) We deleted figure 2C. because we described the oocyte ratios from the ovary and the same data in table 1 which is the percentage data of collected oocytes. Sorry for make confusion.

FIG. 3B, FIG. 3C and FIG. 4 should be quantified so that the differences can be seen more directly

  1. R) Figure 3B is the gene expression pattern of the ovary and Figure 3A is the quantification analysis data of gene expression. Figure 3C is the quantification analysis data of the ovary. And Figure 4 is the LDLR protein expression pattern in the ovary. We present quantification analysis data in Figure 3C.

Please add n=X information to the notes for Figures 2, 3 and 4

  1. R) We updated all of n=x information at figure.

In lines 172-177, whether Figure 5 is verified or conjectured, please add background information for further details

  1. R) We updated the information in the figure. Our figure data present conjectured LDLR signal pathway.

For lines 196-199, please add feeding conditions for mice

  1. R) we follow experiment standard condition of animal maintain with regular feeding

Line 227, 4.5. Gene expression analysis should include PCR and qPCR. I suggest combining them together or modifying the title 4.5

  1. R) PCR and qPCR procedure a little condition and protocols, so we separated both methods.

Lines 240-270, Please add quantitative analysis

  1. R) we added that information regarding mRNA and protein amount analysis methods

Please add full blotting in supplementary material

  1. R) we already added blotting images in the original images data.

The P value of the full text should be italicized

  1. R) Changed italics

More attention should be paid to the superscript and subscript of the manuscript and annex text, such as lines 5 and 9 of the annex

  1. R) We have updated and followed the reviewer's comment. Thanks again for your valuable comment for us.

Reviewer 3 Report

Authors investigate the effect of statin administration prior to superovulation on follicular development and expression of LDLR and other molecules that are related to steroidogenesis in mice. As a complex biological system, a better understanding of the follicular microenvironment would undoubtedly contribute to actual knowledge on mammalian folliculogenesis. The basis of the presented study is thus very interesting. Nonetheless, the methodology lacks rigor and the experimental design cannot adequately answer the experimental questions presented by authors. Improvements to the methodological design are thus essential and need to be revised. In addition, more evidence is required to substantiate arguments in both the introduction and discussion, as well as for the validity of the presented results. There are also a few language and editing errors  that need to be addressed in the text.   Specific comments:  

L19 Please specify hormone used to superovulate in the abstract

L47 A word appears to be missing from the sentence. Please revise

L53-54  Sentence lacks clarity -low expression? high expression? Please specify.

L 73-79 Supporting evidence that statins can upregulate LDLR is lacking in the introduction. No references are presented in this paragraph.

L85-90 Why was viability of cells after lovastatin administration assessed and why were HEK293 cells chosen as a model? Was the purpose to establish the dose to be administered in vivo? Rationale?

L92 In this sentence it appears that lovasatin was injected to mice for ten consecutive days whilst  the methodology states that a single injection was given 10 days prior to superovulation. Please clarify.

L93-94 The indicated age for start of treatment here is “postnatal d21-28". This is inconsistent with what is declared in the Methods section on L 196: “Outbred white ICR mice (6–8 weeks old…” were used. This may impact the results, since in the first group mice may have not reached puberty, whilst in the second they are pospuberal.

L105 declares the number of “collected oocytes”. There is no mention in the Methods section of the procedure followed to collect oocytes or how they were evaluated.

L106-109 If differences are significant please state clearly and add p values

L124 Authors declare that “Lovastatin treatment also increased LDLR expresion”, however no values are presented in the text to support this claim, and no statistical difference is indicated in figure 3A. The same is true for StAR and AMH. Please clarify.

L136-137 Please specify results and p values to support differences between groups.

L 145 Authors indicate “assessing follicular development and in vitro maturation of oocytes”. There is no evidence of performing culture of oocytes in this study. Nor is there an explanation of criteria used for classification. Please clarify.

L145-147 No evidence is shown to support these statements by the authors. The comparisons between stages of follicular development are not properly presented in this study. The methodology as submitted is not thought out to answer this question. To declare differences between gonadotropin independent and dependent stages of follicular growth authors mention the upregulation of  pluripotency markers related to  “primordial germ cells” in L152-153. However, this statement cannot substitute considerations of an appropriate experimental design to exert such claims. Moreover,  albeit there is controversy around the existence of primordial germ cells in females, most evidence continues to support the lack of these cells in the female ovary after birth.

L156 Again, are there in fact primordial germ  cells in “females of advanced age”, and if so, would their number be large enough to yield the results presented by the authors? Can the results be derived from the study with the presented methodology? Are these the only cells to express these markers, considering that cell populations of the ovary were not separated?

L166 No statistical evidence of upregulation of AMH is clearly presented in text or figures (see comments for figures below).

L202 Reasoning for the time of Lovastatin injection and for superovulation is lacking.

L204 When were the mice euthanized exactly? Hours, days after eCG injection? Corpora lutea appear to be present in histological images. When were ovaries harvested in relation to the start of superovulation? Did the animals ovulate? Figure 4 seems to show ovaries containing corpora lutea.

L212 Please change the word “testicular” to “ovarian.”

L272-273 Authors state that “all experiments were repeated more than three times”. How many times exactly? How many animals and ovaries were used for these repetitions. How was this accounted for in the statistical analyses? Simply performing one-way ANOVA would not be accurate.

Figures 1A and 1B reasoning behind presenting the results of Body weight changes of lovastatin-treated mice are lacking in the manuscript. Both figures present the same data, keep just one.

Figures 2b and 2c Are there statistical differences between groups? They are not shown in the graphs, but a difference is stated in the text preceding the image

Fig 4. When and how many ovaries were collected for each experiment?. What are they showing? Are there CL present? Was ovulation spontaneous or induced? If it is the latter, how was it induced?

Table 1 There is no mention in the Methods section of the procedure followed to collect oocytes or how they were evaluated. From how many ovaries from each group derive these results? Are there statistical differences? 

Additional comments:  

HDL are the only lipoproteins capable of diffusing through the follicular basal lamina because larger particles are excluded due to the size of its pores. The interest in intrafollicular LDLR needs to be better substantiated by the authors.

Statistical analysis. There is no description of an experimental design for many of the variables studied (follicle number, gene or protein expression, etc). How many animals were used for the study of each variable? If the study of each variable encompassed at least 3 experiments, how many mice were included per treatment in each experiment? How was the variability between animals within and between experiments considered to test statistical differences?  

Authors mention collection of oocytes at different developmental stages and in vitro culture. No mention of these procedures are found in the methods section. Which and how many ovaries were used? 

Half-life of lovastatin is 1.1 to 1.7h. How long did the pharmacological effect of lovastatin last for? Was the effect still present in mice at superovulation? Or do the authors suggest that the effect of statin is exerted in the follicles in early stages of development and expressed later at superovulation? Discuss.

Did the authors measure the effect of lovastatin on cholesterol serum concentration? How did treatment affect circulating HDL, LDL and VLDL?

Author Response

Reviewer3#

Comments and Suggestions for Authors

Authors investigate the effect of statin administration prior to superovulation on follicular development and expression of LDLR and other molecules that are related to steroidogenesis in mice. As a complex biological system, a better understanding of the follicular microenvironment would undoubtedly contribute to actual knowledge on mammalian folliculogenesis. The basis of the presented study is thus very interesting. Nonetheless, the methodology lacks rigor and the experimental design cannot adequately answer the experimental questions presented by authors. Improvements to the methodological design are thus essential and need to be revised. In addition, more evidence is required to substantiate arguments in both the introduction and discussion, as well as for the validity of the presented results. There are also a few language and editing errors  that need to be addressed in the text.  

 Specific comments:  

L19 Please specify hormone used to superovulate in the abstract

  1. R) Add it

L47 A word appears to be missing from the sentence. Please revise

  1. R) revise it.

L53-54  Sentence lacks clarity -low expression? high expression? Please specify.

  1. R) low expression. Add it.

L 73-79 Supporting evidence that statins can upregulate LDLR is lacking in the introduction. No references are presented in this paragraph.

  1. R) we added the reference

L85-90 Why was viability of cells after lovastatin administration assessed and why were HEK293 cells chosen as a model? Was the purpose to establish the dose to be administered in vivo? Rationale?

  1. R) Until now there is mouse model study with statins. So we try to identified no toxic ranges of lovastatins. We try to determine optimal dose with cells for in vivo study.

L92 In this sentence it appears that lovasatin was injected to mice for ten consecutive days whilst  the methodology states that a single injection was given 10 days prior to superovulation. Please clarify.

  1. R) we updated the methodology in the text

L93-94 The indicated age for start of treatment here is “postnatal d21-28". This is inconsistent with what is declared in the Methods section on L 196: “Outbred white ICR mice (6–8 weeks old…” were used. This may impact the results, since in the first group mice may have not reached puberty, whilst in the second they are pospuberal.

  1. R) Sorry to make confusion We have put in the wrong day. We treated a 7 ~ 9-week-old mouse which is postnatal 28 ~ 36 days. We purchased 6~8 weeks mice and one week keeping the animal house then we started the experiment.

L105 declares the number of “collected oocytes”. There is no mention in the Methods section of the procedure followed to collect oocytes or how they were evaluated.

  1. R) we updated the methodology in the text

L106-109 If differences are significant please state clearly and add p values

  1. R) we put in the p values on the significant data.

L124 Authors declare that “Lovastatin treatment also increased LDLR expresion”, however no values are presented in the text to support this claim, and no statistical difference is indicated in figure 3A. The same is true for StAR and AMH. Please clarify.

  1. R) LDLR no significantly different between control and experiment at the gene level. But protein level, lovastatin shows significantly enhanced in the experiment group. Practically, protein is a biological function factor compared to the gene level.

L136-137 Please specify results and p values to support differences between groups.

  1. R) We marked the p value

L 145 Authors indicate “assessing follicular development and in vitro maturation of oocytes”. There is no evidence of performing culture of oocytes in this study. Nor is there an explanation of criteria used for classification. Please clarify.

  1. R) thank you for your comment, we delete the words from text.

L145-147 No evidence is shown to support these statements by the authors. The comparisons between stages of follicular development are not properly presented in this study. The methodology as submitted is not thought out to answer this question. To declare differences between gonadotropin independent and dependent stages of follicular growth authors mention the upregulation of  pluripotency markers related to  “primordial germ cells” in L152-153. However, this statement cannot substitute considerations of an appropriate experimental design to exert such claims. Moreover,  albeit there is controversy around the existence of primordial germ cells in females, most evidence continues to support the lack of these cells in the female ovary after birth.

  1. R) we have missed explain. That is primary oocyte which is gonadotrophin independent phase. We re-writing primary oocyte instead of primordial germ cells.

L156 Again, are there in fact primordial germ  cells in “females of advanced age”, and if so, would their number be large enough to yield the results presented by the authors? Can the results be derived from the study with the presented methodology? Are these the only cells to express these markers, considering that cell populations of the ovary were not separated?

  1. R) it is same wrong word issue in the text. Also we re-writing in the text. sorry

L166 No statistical evidence of upregulation of AMH is clearly presented in text or figures (see comments for figures below).

  1. R) updated the sentence in the text.

L202 Reasoning for the time of Lovastatin injection and for superovulation is lacking.

  1. R) we want to observation effect of lovastatin as support from primary oocyte to follicle development.

L204 When were the mice euthanized exactly? Hours, days after eCG injection? Corpora lutea appear to be present in histological images. When were ovaries harvested in relation to the start of superovulation? Did the animals ovulate? Figure 4 seems to show ovaries containing corpora lutea.

  1. R) we didn’t inject hCG. We just inject PMSG for follicle development phase. Because in clinical, poor ovarian responder and poor ovarian failure have a issue at the recruit of primary oocyte in the ovary. Therefore we want to see lovastatin enhanced number of primary oocyte and follicle development in the ovary.

L212 Please change the word “testicular” to “ovarian.”

  1. R) Re-writing that word. Very sorry miss-writing the text

L272-273 Authors state that “all experiments were repeated more than three times”. How many times exactly? How many animals and ovaries were used for these repetitions. How was this accounted for in the statistical analyses? Simply performing one-way ANOVA would not be accurate.

  1. R) 3 times repeat the full set experiment (We used ten mice as control and ten mice as an experiment for one set). We asked statistical experts, and they recommended one-way ANOVA methods for our experiment data.

Figures 1A and 1B reasoning behind presenting the results of Body weight changes of lovastatin-treated mice are lacking in the manuscript. Both figures present the same data, keep just one.

  1. R) We delete Fig 1A and keeping B.

Figures 2b and 2c Are there statistical differences between groups? They are not shown in the graphs, but a difference is stated in the text preceding the image

  1. R) we updated figure 2. We delete figure 2c

Fig 4. When and how many ovaries were collected for each experiment?. What are they showing? Are there CL present? Was ovulation spontaneous or induced? If it is the latter, how was it induced?

Table 1 There is no mention in the Methods section of the procedure followed to collect oocytes or how they were evaluated. From how many ovaries from each group derive these results? Are there statistical differences? 

  1. R) we used control group 30 and experiment group 30 mouse. Then half ovary harvest oocyte which is 30 ovaries from each group. Then other half group 30 ovaries, we apply for histology, gene expression, and protein expression analysis.

Additional comments:  

HDL are the only lipoproteins capable of diffusing through the follicular basal lamina because larger particles are excluded due to the size of its pores. The interest in intrafollicular LDLR needs to be better substantiated by the authors.

  1. R) Thank you very much for your comment. We prepared next study for aging model with statin. We try to analysis deeper in the ovary like intrafollicular LDLR. Thank you again.

Statistical analysis. There is no description of an experimental design for many of the variables studied (follicle number, gene or protein expression, etc). How many animals were used for the study of each variable? If the study of each variable encompassed at least 3 experiments, how many mice were included per treatment in each experiment? How was the variability between animals within and between experiments considered to test statistical differences?  

  1. R) we updated the experimental group number and information for study. We analysis by ANOVA one methods for experiment data statistically difference or not. Software recommends this methods.

Authors mention collection of oocytes at different developmental stages and in vitro culture. No mention of these procedures are found in the methods section. Which and how many ovaries were used? 

  1. R) we up-dated information and re-writing in text regarding number of ovaries

Half-life of lovastatin is 1.1 to 1.7h. How long did the pharmacological effect of lovastatin last for? Was the effect still present in mice at superovulation? Or do the authors suggest that the effect of statin is exerted in the follicles in early stages of development and expressed later at superovulation? Discuss.

  1. R) Very interesting comment for us. We didn’t think about that. Now we don’t have answer. But we are moving the next study with old mouse. I will check-up the more precise effect analysis depend on the follicle phase.

Did the authors measure the effect of lovastatin on cholesterol serum concentration? How did treatment affect circulating HDL, LDL and VLDL?

  1. R) this experiment focus on the LDLR in the ovary. Your comment interesting for me. We further study regarding other factor like HDL, LDL and VLDL.

Reviewer 4 Report

1. Abbreviation is missing in Table 1.

2. Authors need to modify the tenses used in the manuscript.

Author Response

Reviewer4#

Comments and Suggestions for Authors

Abbreviation is missing in Table 1.

  1. R) add it.
  2. Authors need to modify the tenses used in the manuscript.
  3. R) I’m sorry missing the table 1. And we re-edited with native speaker. Thank for your comment.

Round 2

Reviewer 3 Report

I appreciate the authors' prompt response, and again aknowledge that the basis of the presented study is very interesting. Nonetheless, the comments made on the first review need to be more carefully adressed and the indication regarding substance in introduction and discussion, as well as flaws in experimental design and description still stand in my view. Moreover these weaknesses question the soundness of drawn results. The number of ovaries used for each phase of the 3 (or 4?) experiments are still not clear for example, nor is the technique used to classify stages of development of oocytes (GV, GVBD,  MII). Also variability within and between experimments needs to be accounted for in the statistical analysis (to name a few of the issues that need to be resolved). I would encourage the authors to continue their important work, but believe that precisely for this reason it is imperative that data and results are presented with reliability and solidity. 

Author Response

Reviewer 3#

Comments and Suggestions for Authors

I appreciate the authors' prompt response, and again aknowledge that the basis of the presented study is very interesting.

Nonetheless, the comments made on the first review need to be more carefully addressed and the indication regarding substance in introduction and discussion, as well as flaws in experimental design and description still stand in my view.

Moreover these weaknesses question the soundness of drawn results.

The number of ovaries used for each phase of the 3 (or 4?) experiments are still not clear for example, nor is the technique used to classify stages of development of oocytes (GV, GVBD, MII).

R) Sorry we make a confusing for you. We have updated the missing part of oocyte handling in the text. We explain methods regarding the oocyte experiment. We collected total 15 ovaries from the control and experiment groups. Therefore, we only inject PMSG (without hCG) which means we harvest immature GV-stage of oocytes from each group. Then we try to in vitro maturation of GV oocytes and analyzed in vitro maturation ratios into GVBD and metaphase II oocytes. We have written in the manuscript regarding in vitro maturation protocols. Also we re-written data in the results and table 1.

Also variability within and between experimments needs to be accounted for in the statistical analysis (to name a few of the issues that need to be resolved).

I would encourage the authors to continue their important work, but believe that precisely for this reason it is imperative that data and results are presented with reliability and solidity.

R) thank you for your positive comment for us.